# Effects of Chronic Diseases on All-Cause Mortality in People with Mental Illness: A Retrospective Cohort Study Using the Korean National Health Insurance Service-Health Screening

**DOI:** 10.3390/ijerph19169989

**Published:** 2022-08-13

**Authors:** Sujin Son, Yun Jin Kim, Seok Hyeon Kim, Johanna Inhyang Kim, Sojung Kim, Sungwon Roh

**Affiliations:** 1Department of Psychiatry, College of Medicine, Hanyang University, 222 Wangsimni-ro, Seongdong-gu, Seoul 04763, Korea; 2Department of Psychiatry, Hanyang University Hospital, 222-1 Wangsimni-ro, Seongdong-gu, Seoul 04763, Korea; 3Biostatistical Consulting and Research Lab, Medical Research Collaborating Center, Hanyang University, 222 Wangsimni-ro, Seongdong-gu, Seoul 04763, Korea

**Keywords:** mental illness, chronic disease, mortality, health screening, retrospective cohort study

## Abstract

The aim of this study was to compare mortality and the prevalence of chronic diseases between people with mental illness and the general population, and to explore which chronic diseases increase the risk of all-cause mortality, especially in people with mental illness. This study assessed data from the 2002–2019 Korean National Health Insurance Service-Health Screening sample cohort. Results revealed that all-cause mortality was higher in people with mental illness compared to people without mental illness (11.40% vs. 10.28%, *p* = 0.0022). Several chronic diseases have a higher prevalence and risk of all-cause mortality in individuals with mental illness than the general population. Among people with the same chronic disease, those with mental disorders had a higher risk of all-cause mortality. Cancer (aHR 2.55, 95% CI 2.488–2.614), liver cirrhosis (aHR 2.198, 95% CI 2.086–2.316), and arrhythmia (aHR 1.427, 95% CI 1.383–1.472) were the top three chronic diseases that increased the risk of all-cause mortality in people with mental illness compared to people without mental illness. Our results suggest the need for more attention to chronic diseases for people with mental illness in clinical practice by explaining the effect of chronic disease on all-cause mortality in people with mental illness.

## 1. Introduction

People with mental illness have a high mortality rate. As has been found in a variety of previous studies, the mortality rate for severe mental illness was more than twice the mortality rate for individuals in a comparable general population [1,2,3,4]. Several studies have reported that the risk of all-cause death for people with mental disorders is two to three times higher than for the general population [5,6]. A study of individuals in the Nordic countries found that men and women with mental disorders lived 20 and 15 years less than the general population, respectively [7]. Some of the excess mortality was due to suicide or accidents, but a large proportion of mortality was due to physical illness. A global review of causes of death reported that 67.3% of people with mental illness died from natural causes [8]. Most people with mental illness died of natural causes such as cardiovascular diseases, cancer, and respiratory diseases [9]. Among natural cause of death, deaths from chronic diseases are preventable deaths in individuals with mental disorders [4]. In addition, comorbid chronic diseases in people with mental illness are associated with an elevated symptom burden and decreased length and quality of life [10]. Unfortunately, the prevalence of chronic diseases among patients with mental illness is higher than in the general population. A meta-analysis of the comorbidities of mental disorders and chronic physical diseases reported that the pooled prevalence of mental disorders in patients with chronic physical diseases was 36.6% (95% CI, 31.4–42.1), and the pooled odds ratio was 3.1 (95% CI, 1.7–5.2) [11]. Many studies have examined the association between mental disorders and chronic diseases such as cancer [12], heart disease [13,14], stroke [15], diabetes [16,17,18], chronic obstructive pulmonary disease (COPD) [19,20], and chronic liver disease [21]. However, large-scale cohort studies, which include various chronic diseases as risk factors for mortality in patients with psychiatric illness compared with the general population, have not yet been conducted.

In this study, we hypothesized that certain chronic diseases increase the risk of all-cause mortality in people with mental illness compared to the general population. This study aimed to compare mortality and the prevalence of chronic diseases between people with mental illness and the general population, and to explore which chronic diseases increase the risk of all-cause mortality, especially in people with mental illness, using data from the Korean National Health Insurance Service-Health Screening (KNHIS-HEALS).

## 2. Methods

### 2.1. Study Population

In this study, we used data from the KNHIS-HEALS cohort. The KNHIS is public health insurance that provides universal health coverage to almost all Koreans. The KNHIS offers the National Screening Program, a biennial health screening program, to all people over 40 years of age. To construct the KNHIS-HEALS database, a sample cohort was selected from among the 2002 and 2003 National Screening Program participants, who were aged between 40 and 79 years in 2002, and followed up through 2019 [22]. This cohort included 513,655 people, comprising a 10% simple random sample of all National Screening Program participants. From 2002 to 2019, information on qualifications and income (socioeconomic variables), hospital and clinic usage history, codes from the clinically determined International Classification of Diseases, 10th revision (ICD-10), and National Screening Program results were constructed into a cohort. 

This study was approved by the Hanyang University Institutional Review Board (IRB number: HYU-2021-097). The authors followed the guidelines of the Declaration of Helsinki (1975). The requirement for informed consent was exempted because the KNHIS database was constructed after anonymization, according to strict confidentiality guidelines.

In this study, from the 513,655 people who comprised the National Screening Program sample, 493,164 participants were selected; 20,491 people who died between 1 January 2002 and 31 December 2009 were excluded. The classification of people with mental illnesses in this study was based on the diagnostic criteria in the ICD-10—codes F00–F99. People with mental illnesses who had a record of being hospitalized, had been outpatients, or were admitted to the day ward at a medical health institution at least once with an F code diagnosis as a major illness between 1 January 2002 and 31 December 2009 were classified as people with mental illness in this study. Regardless of the presence or absence of a psychiatric diagnosis before 2002, people who had psychiatric medical records between 2002 and 2009, were defined as people with mental illness. A total of 7199 people with mental illness and 485,965 people without mental illness were identified. The observation period was from 1 January 2010 to 31 December 2019.

### 2.2. Measurements

Gender, age, and income level were based on 2002. The number of health screenings was the total number of health screenings between 2002–2009. Since the study population was selected from the 2002 and 2003 National Screening Program participants, who were aged between 40 and 79 years in 2002, age was divided into 40–50, 50–59, 60–69, and more than 70 years. The income level was divided into the 0–3 decile, the 4–7 decile, and the 8–10 decile. The number of health screenings was divided into 0, 1, 2, 3, 4, and 5. Physical activity, smoking, and alcohol drinking were based on the last health screenings during 2002–2009. According to the recommended physical activity standards provided by the KNHIS-HEALS, poor physical activity was defined as exercising moderate-intensity physical activity less than 5 days a week and high-intensity physical activity less than 3 days a week. Smoking was categorized as never smoking, ex smoking, or current smoking. Referring to the high-risk drinking standards of the Korean National Health and Nutrition Examination Survey, alcohol drinking was categorized as non-drinking, drinking less than twice a week, or drinking more than twice a week. To simplify the category, the amount of alcohol consumed per serving was not considered. Blood pressure, total cholesterol, and fasting blood glucose were averages from 2002–2009. Total cholesterol (mg/dL) was categorized into 3 groups: <200 mg/dL (normal), 200–239 (borderline high), and ≥240 mg/dL (high) [23]. Blood pressure (mmHg) was categorized into 3 groups: systolic blood pressure (SBP) < 120 and diastolic blood pressure (DBP) < 80 (normal), SBP 120–139 or DBP 80–89 (prehypertension), and SBP > 140 or DBP > 90 (hypertension). Fasting glucose sugar level (mg/dL) was categorized into 3 groups: ≤99 (normal), 100–125 (prediabetes), and ≥126 (diabetes). The classification of chronic diseases was based on a list of 20 chronic diseases selected by the U.S. Office of the Assistant Secretary of Health [23]. Among them, five mental disorders (autism spectrum disorder, dementia, depression, schizophrenia, and substance abuse disorders) were excluded because they could be included in duplicate in the case of people with mental illness. Additionally, AIDS–HIV was excluded because there was no case in this study. In addition, liver cirrhosis, which is common in Korea, was included. Chronic diseases were classified according to the ICD-10 codes for diagnosis at the time of outpatient or inpatient treatment, based on medical records during the observation period. Therefore, in total, 15 chronic diseases were selected, as follows: hypertension (I10–I15), congestive heart failure (I50), coronary artery disease (I20–125), arrhythmia (I44–I49), hyperlipidemia (E78), stroke (I60–69), arthritis (M00–M25), asthma (J45–J46), cancer (C00–C97), chronic kidney disease (N18–N19), COPD (J44, J47), diabetes (E10–E14), chronic hepatitis (B18), liver cirrhosis (K702, K703, K746), and osteoporosis (M81). In case of death during the follow-up period from 1 January 2010 to 31 December 2019, the survival period until the date of death was calculated. On the other hand, in the case of no death during the follow-up period, the survival period was calculated until the time censored (31 December 2019). Information on death (date and cause) from Statistics Korea was individually linked using unique personal identification numbers. Based on this, the risk of all-cause mortality for people with and without mental illness was calculated.

### 2.3. Statistical Analysis

Independent t-tests were used to compare normally distributed variables, and Wilcoxon’s rank-sum test was used to compare non-normally distributed variables. The chi-square and Fisher’s exact test were used to compare categorical variables. Kaplan–Meier survival curves were used to compare the survival rates of people with mental illness versus those without mental illness. The Cox proportional hazards model was used to investigate the association between chronic diseases and risk of all-cause mortality. The results are presented as adjusted hazard ratios (aHRs) with 95% confidence intervals (95% CIs) to analyze risk of mortality. All Cox proportional hazards models were fully adjusted for age, sex, income level, the number of health screenings, physical activities, smoking, alcohol consumption, body mass index, blood pressure, fasting blood glucose levels, total cholesterol, LDL, and chronic diseases. All analyses were conducted using the SAS software (version 9.4; SAS Institute, Cary, NC, USA). Statistical significance was determined using a two-tailed test, with a *p*-value of 0.05 as the threshold.

## 3. Results

A total of 493,164 participants were included in the analyses—7199 (1.46%) were people with mental illness and 485,965 (98.54%) were people without mental illness. Of the participants, 50,995 (821 with mental illness and 49,974 without mental illness) died during the study period. The percentage of deaths from all causes was higher among people with mental illness than those without mental illness (11.40% vs. 10.28%, *p* = 0.0022). Figure 1 shows the Kaplan–Meier survival curve for people with and without mental illness. People with mental illness showed a lower survival probability than those without mental illness. Compared to people without mental illness, people with mental illness had a higher risk of all-cause mortality (HR 1.089, 95% CI 1.017–1.167). 

Table 1 compares the differences in sociodemographic and clinical characteristics between the people with and without mental illness. People with mental illness had a lower number of health screenings than people without mental illness (*p* < 0.0001). Gender, age, smoking, alcohol drinking, blood pressure, total cholesterol, and fasting glucose level were also statistically significant differences between people with and without mental illness. 

Table 2 presents that prevalence of chronic diseases in people with mental illness and people without mental illness. Compared to people without mental illness, people with mental illness had a higher proportion of chronic diseases, as follows: hypertension (52.06% vs. 49.39%, *p* < 0.0001), coronary artery disease (18.03% vs. 16.15%, *p* < 0.0001), arrhythmia (7.74% vs. 6.63%, *p* = 0.0002), hyperlipidemia (53.55% vs. 49.58% *p* < 0.0001), stroke (17.68% vs. 14.39% *p* < 0.0001), arthritis (68.05% vs. 63.68%, *p* < 0.0001), asthma (22.79% vs. 20.86%, *p* < 0.0001), COPD (7.51% vs. 6.87%, *p* = 0.0327), and osteoporosis (26.46% vs. 21.81%, *p* < 0.0001). 

Table 3 presents multivariable Cox proportional hazard regression analysis of the association of chronic diseases and risk of all-cause mortality in people with and without mental illness. Among chronic diseases, hypertension (aHR 1.394, 95% CI 1.089–1.785 vs. aHR 1.192, 95% CI 1.158–1.228), congestive heart failure (aHR 1.399, 95% CI 1.048–1.866 vs. aHR 1.219, 95% CI 1.177–1.263), cancer (aHR 2.846, 95% CI 2.310–3.506 vs. aHR 2.547, 95% CI 2.484–2.611), and liver cirrhosis (aHR 2.365, 95% CI 1.472–3.8 vs. aHR 2.196, 95% CI 2.084–2.315) had higher risk of all-cause mortality in people with mental illness compared to people without mental illness. 

Table 4 shows multivariable Cox proportional hazard regression analysis results for the effects of mental illness on the risk of all-cause mortality among people with chronic diseases. Among people with the same chronic disease, those with mental disorders had a higher risk of all-cause mortality. Compared to people without mental illness but with chronic diseases, people with mental illness and chronic diseases had higher risk of mortality in 11 chronic diseases; in the following order—cancer (aHR 2.55, 95% CI 2.488–2.614), liver cirrhosis (aHR 2.198, 95% CI 2.086–2.316), arrhythmia (aHR 1.427, 95% CI 1.383–1.472), chronic kidney disease (aHR 1.363, 95% CI 1.304–1.423), chronic obstructive pulmonary disease (aHR 1.313, 95% CI 1.274–1.353), stroke (aHR 1.31, 95% CI 1.277–1.344), diabetes (aHR 1.251, 95% CI 1.218–1.285), CHF (aHR 1.222, 95% CI 1.18–1.265), hypertension (aHR 1.195, 95% CI 1.16–1.23), chronic hepatitis (aHR 1.131, 95% CI 1.024–1.25), and asthma (aHR 1.088, 95% CI 1.06–1.116). However, people with mental illness and arthritis (aHR 0.856, 95% CI 0.834–0.880), hyperlipidemia (aHR 0.807, 95% CI 0.786–0.829), or osteoporosis (aHR 0.913, 95% CI 0.888–0.939) had a lower risk of all-cause mortality than those without a mental disorder. CAD did not have a significant difference in the risk of all-cause mortality between people with mental illness and without mental illness (aHR 1.005, 95% CI 0.978–1.033).

To differentiate between the severity of mental illness, we divided the participants into people with history of psychiatric hospitalization, people with history of psychiatric outpatient only, and people without mental illness. The number of people with history of psychiatric hospitalization was 404 (0.08%), and the number of people with a history of psychiatric outpatient treatment only was 6795 (1.38%). The all-cause mortality of people with a history of psychiatric hospitalization was higher than those of people with a history of psychiatric outpatient only, and those of people without mental illness (25.00% vs. 10.60% vs. 10.28%, *p* < 0.0001). The results of the sub-analysis are as follows. 

Table 5 presents the sociodemographic and clinical characteristics between three groups. In particular, people with a history of psychiatric hospitalization had a lower number of health screenings than people with a history of psychiatric outpatient and people without mental illness (*p* < 0.0001). Gender, age, smoking, alcohol drinking, blood pressure, total cholesterol, and fasting glucose level were also statistically significant differences between three groups.

Table 6 presents the prevalence of chronic diseases between three groups. Compared to people with a history of psychiatric outpatient care and people without mental illness, people with a history of psychiatric hospitalization had a higher proportion of chronic diseases, as follows: hypertension (56.93% vs. 51.77% vs. 49.39%, *p* < 0.0001), CHF (7.18% vs. 5.28% vs. 4.90%, *p* = 0.0382), CAD (22.77% vs. 17.75% vs. 16.15%, *p* < 0.0001), arrhythmia (9.90% vs. 7.61% vs. 6.63%, *p* = 0.0002), stroke (24.01% vs. 17.31% vs. 14.39%, *p* < 0.0001), asthma (24.50% vs. 22.69% vs. 20.86%, *p* = 0.0002), CKD (3.71% vs. 1.87% vs. 2.12%, *p* = 0.0302), COPD (9.65% vs. 7.39% vs. 6.87%, *p* = 0.0210), diabetes (42.08% vs. 30.76% vs. 30.49%, *p* < 0.0001), and liver cirrhosis (2.48% vs. 1.02% vs. 1.25%, *p* = 0.0187).

Table 7 presents association of chronic diseases and risk of all-cause mortality between three groups. Among risk factors, cancer (aHR 4.094, 95% CI 1.474–11.373 vs. aHR 2.820, 95% CI 2.270–3.502 vs. aHR 2.547, 95% CI 2.484–2.611) and liver cirrhosis (aHR 26.334, 95% CI 3.685–188.178 vs. aHR 2.160, 95% CI 1.291–3.613 vs. aHR 2.196, 95% CI 2.084–2.315) showed a higher aHR in people with a history of psychiatric hospitalization than people with a history of psychiatric outpatient and people without mental illness.

## 4. Discussion

This study aimed to compare mortality and the prevalence of chronic diseases between people with mental illness and the general population, and to explore which chronic diseases increase the risk of all-cause mortality especially in people with mental illness. We found that all-cause mortality of people with mental illness is higher than that of people without mental illness. People with mental illness had a higher prevalence of hypertension, coronary artery disease, arrhythmia, hyperlipidemia, stroke, arthritis, asthma, COPD, and osteoporosis than people without mental illness. Among chronic diseases, cancer, liver cirrhosis, and arrhythmia were the top three chronic diseases that increased the risk of all-cause mortality in people with mental illness compared to people without mental illness. The sub-analysis showed that the difference between people history of psychiatric hospitalization and those with history of psychiatric outpatient was greater than the difference between those with a history of psychiatric outpatient and those without mental illness.

To the best of our knowledge, this is the first study to reveal an association between multiple chronic diseases and the risk of all-cause mortality in people with and without mental illness in a nationwide sample. Compared with previous studies, in this study, the difference in mortality between people with mental illness and the general population was rather small. In this study, we extracted and analyzed information on patients with various mental disorders from all the hospitals and clinics that served the population of South Korea. We defined people with mental illness as those with ICD-10 F-code diagnoses, regardless of the level of institutionalization or symptom severity. Therefore, our populations included many mild patients who did not did not differ significantly from the general population. Nevertheless, the result showed that mortality was higher among people with a wide range of mental disorders than those among the general population. 

Our results showed that the prevalence of chronic cardiovascular diseases, such as hypertension, coronary artery disease, arrhythmia, hyperlipidemia, and stroke, among people with mental illness is much higher than that of the general population. The high prevalence of cardiovascular diseases in people with mental illness is consistent with previous studies on population in other countries [24,25,26]. In this study, arrhythmia, congestive heart failure, and hypertension increased the risk of all-cause mortality in people with mental illness. In addition, patients with mental illness and with hypertension, congestive heart failure, or arrhythmia had a higher risk of all-cause mortality than those with each of these diseases but no mental illness. These results might be explained by the high proportion of risk factors for cardiovascular disease, low treatment rate, and use of antipsychotic drugs. Several studies have reported that unhealthy lifestyles, such as smoking, lack of exercise, and an unhealthy diet, increase the prevalence of cardiovascular disease and risk of death from cardiovascular disease in people with mental illness [27,28]. A study on the effect of cardiac care deficiency on the high mortality of patients with schizophrenia indicated that the quality of medical treatment provided to patients with schizophrenia with cardiac conditions is often suboptimal, and may be linked to avoidable excess mortality [29]. People with mental illness are less likely to have their weight or blood pressure measured in primary care [30], or to be assessed or treated for hyperlipidemia [31,32]. Evidence from controlled studies, including large-scale randomized trials, indicates that some, but not all, antipsychotic drugs can adversely affect adiposity, as well as glucose and lipid metabolism [33,34,35]. These considerations contribute to the risk of cardiovascular disease and mortality in people with mental illness.

In this study, there was no significant difference in the prevalence of cancer between people with and without mental illness; however, patients with cancer and mental illness had a risk of all-cause mortality that was more than two times higher than patients with cancer but without mental illness. These results could be interpreted as a result of poor diagnosis and treatment of cancer in people with mental illness. A study of cancer-related mortality in people with mental illness in Western Australia found that people with mental illness had a lower prevalence of cancer than the general population, but had a higher mortality rate because they were more likely to have metastases at diagnosis and less likely to receive specialized interventions [36]. This suggests that more intense screening and treatment for cancer is needed to reduce the cancer related mortality of people with mental illness.

Liver cirrhosis also significantly affects the risk of all-cause mortality, with an increase of over two times in psychiatric patients compared to non-psychiatric patients. These results could be explained by excessive alcohol consumption by people with mental illness [37]. Alcohol use disorder is the main cause of liver cirrhosis, and it has a high mortality rate. A study of mental–physical comorbidity and mortality identified that alcohol use disorder was associated with the highest odds of comorbid chronic physical illness and five-year mortality [38]. Because this study did not include liver disease, the contribution of alcohol use disorder to the risk of death from liver disease was not clear. However, a study of alcohol use disorder and associated chronic diseases from France reported that the elevated risk of death was highest for liver diseases, which were associated with approximately two-third of deaths in patients with alcohol use disorders [39]. Furthermore, the most common etiologies of cirrhosis were alcohol-related liver disease and hepatitis C infection [40]. Considering that injections for drug use are the main cause of hepatitis C, alcohol use disorders, as well as other substance use disorders, account for the high risk of death-related cirrhosis in patients with mental illness. In addition, some psychiatric medications, such as neuroleptics, mood stabilizers, and a few antidepressants, are associated with drug-induced liver injury [41]. 

## 5. Strengths and Limitations

This study had several limitations. Above all, there were clear limitations because of the data of this study from the KNHIS-HEALS cohort database. In the KNHIS-HEALS cohort database, mental illness is classified as a sensitive disease, so detailed diagnosis names are not indicated, and all are indicated as F***. As a result, it was not possible to describe the distribution of diagnoses of mental disorders. In addition, under the age of 40, only householders of locally provided policyholders or employer-provided policyholders are eligible subjects of the national health screening program. As a result, participants under the age of 40 were not included in this study, because the KNHIS-HEALS cohort database is sampled with subjects over the age of 40, which is the age when all Koreans are eligible for the national health screening program. Furthermore, the KNHIS-HEALS cohort database did not include medical records before 2002 and information on the date of initial diagnosis. As a result, there was a possibility of misclassification of people with mental illness, because people who diagnosed before 2002 but had no psychiatric record between 2002 and 2009 were not included in the category for people with mental illness. Despite such disadvantages, we used the KNHIS-HEALS cohort database because the data included information such as physical activity, smoking, alcohol drinking, blood pressure, fasting glucose level, etc. Since the risk factors could influence mortality, we controlled the risk variables and presented an association between chronic diseases and the risk of all-cause mortality. Future studies to investigate the effects of chronic diseases on all-cause mortality for each psychiatric disorder are proposed. In addition, participants should not be affected by the age of onset for mental disorders, including all ages, and the date of initial diagnosis should be included.

In addition, there is a possibility that the observation period of 10 years was rather short to examine the effect on mortality. As a matter of fact, the fact that the number of people with mental illness was only 1.46% of the total participants and the difference in the number of people with and without mental illness are significant. Additionally, unequal gender and age distribution in the study groups could affect the prevalence of chronic diseases and mortality. These problems could be solved by controlling gender and age through propensity score matching, but this was not conducted due to the disadvantage of reducing the overall sample size.

Despite these limitations, this is the first study to reveal an association between multiple chronic diseases and all-cause mortality in people with mental illness and those without mental illness in a nationwide sample. In particular, we analyzed association with risk of mortality for a total of 15 chronic diseases, which is a strength of this study compared to other studies that include only a few diseases. As a result, we were able to find 11 chronic diseases that increase the risk of all-cause mortality, especially in people with mental illness compared to people without mental illness. These results could be used as a list of chronic diseases that require attention, particularly when treating people with mental illness in clinical practice. In addition, we extracted and analyzed information on patients with all types of mental disorders from all the hospitals and clinics that served the entire population of South Korea. Our study is unique in that we targeted a wider range of mental disorders than previous studies, which targeted only a few mental disorders. 

## 6. Conclusions

People with mental illness had higher all-cause mortality than those without mental illness. Several chronic diseases had a higher prevalence in patients with mental illness than in the general population. Some chronic diseases had a higher risk of all-cause mortality in people with mental illness compared to people without mental illness. Among people with the same chronic disease, those with mental disorders had a higher risk of all-cause mortality. Cancer, liver cirrhosis, and arrhythmia were the top three chronic diseases that increased the risk of all-cause mortality in people with mental illness compared to people without mental illness. These results suggest the need to pay more attention to chronic diseases in people with mental illness in clinical practice by explaining the effect of chronic disease on all-cause mortality in people with mental illness.

## Figures and Tables

**Figure 1 ijerph-19-09989-f001:**
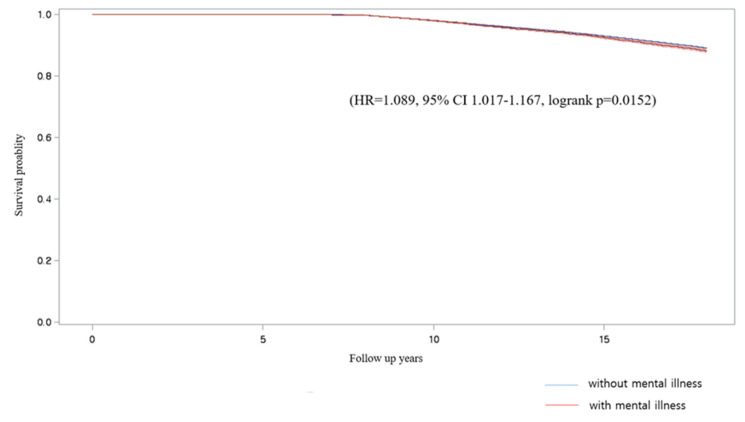
Kaplan–Meier curve for people with mental illness (red) vs. those without mental illness (blue).

**Table 1 ijerph-19-09989-t001:** Comparisons of sociodemographic and clinical characteristics for people with and without mental illness.

Variables	People with Mental Illness (n = 7199)	People without Mental Illness (n = 485,965)	*p*-Value
**Gender**			<0.0001
Male	3195 (44.38)	260,477 (53.60)	
Female	4004 (55.62)	225,488 (46.40)	
**Age**			<0.0001
40–49	3079 (42.77)	225,318 (46.37)	
50–59	2180 (30.28)	141,786 (29.18)	
60–69	1508 (20.95)	93,447 (19.23)	
>70	432 (6.00)	25,414 (5.23)	
**Income level**			0.2225
0–3 decile	1676 (23.28)	109,233 (22.48)	
4–7 decile	2341 (32.52)	158,124 (32.54)	
8–10 decile	3182 (44.20)	218,608 (44.98)	
**Number of health screenings**			<0.0001
≥5	3087 (42.88)	227,135 (46.74)	
4	1490 (20.70)	95,829 (19.72)	
3	802 (11.14)	52,021 (10.70)	
2	577 (8.02)	37,392 (7.69)	
1	561 (7.79)	33,689 (6.93)	
0	682 (9.47)	39,899 (8.21)	
**Physical activity**			0.7705
Good	2168 (33.64)	149,547 (33.82)	
Poor	4277 (66.36)	292,630 (55.18)	
**Smoking**			<0.0001
Never smoking	3224 (49.50)	203,167 (45.56)	
Ex smoking	1090 (16.74)	91,585 (20.54)	
Current smoking	2199(33.76)	151,148(33.90)	
**Alcohol drinking**			<0.0001
None	564 (8.66)	32,014 (7.18)	
<2 times/week	1540 (23.64)	123,539 (27.70	
≥2 times/week	4410 (67.70)	290,373 (65.12)	
**Blood pressure(mmHg)**			0.0055
Normal	1954(29.98)	125,334(28.10)	
Prehypertension	3817(58.57)	267,056(59.87)	
Hypertension	746(11.45)	53,676(12.03)	
**Total cholesterol (mg/dL)**			<0.0001
<200	4836 (74.21)	337,862 (77.25)	
200–239	1354 (20.78)	88,620 (13.33)	
≥240	327 (5.02)	19,584 (4.39)	
**Fasting glucose level (mg/dL)**			<0.0001
<100	3690 (56.62)	249,950 (56.03)	
100–125	2225 (34.14)	146,805 (32.91)	
≥126	602 (9.24)	49,311 (11.05)	

Values are presented as numbers (%).

**Table 2 ijerph-19-09989-t002:** Comparisons of prevalence of chronic diseases for people with and without mental illness.

Chronic Diseases	People with Mental Illness (n = 7199)	People without Mental Illness (n = 485,965)	*p*-Value
Hypertension	3748 (52.06)	240,006 (49.39)	<0.0001
CHF	388 (5.39)	23,827 (4.9)	0.0616
CAD	1298 (18.03)	78,494 (16.15)	<0.0001
Arrhythmia	577 (7.74)	32,212 (6.63)	0.0002
Hyperlipidemia	3855 (53.55)	240,957 (49.58)	<0.0001
Stroke	1273 (17.68)	69,925 (14.39)	<0.0001
Arthritis	4899 (68.05)	309,464 (63.68)	<0.0001
Asthma	1641 (22.79)	101,393 (20.86)	<0.0001
Cancer	745 (10.35)	52,114 (10.72)	0.3183
CKD	142 (1.97)	10,303 (2.12)	0.4096
COPD	541 (7.51)	33,366 (6.87)	0.0327
Diabetes	2260 (31.39)	148,171 (30.49)	0.0989
Chronic hepatitis	62 (0.86)	3691 (0.76)	0.3057
Liver cirrhosis	79 (1.1)	6086 (1.25)	0.2616
Osteoporosis	1905 (26.46)	106,007 (21.81)	<0.0001

CHF: congestive heart failure; CAD: coronary artery disease; CKD: chronic kidney disease; COPD: chronic obstructive pulmonary disease.

**Table 3 ijerph-19-09989-t003:** Association of chronic diseases and risk of all-cause mortality in people with and without mental illness.

	People with Mental Illness		People without Mental Illness	
	aHR	95% CI	aHR	95% CI
Hypertension	1.394	1.089–1.785	1.192	1.158–1.228
CHF	1.399	1.048–1.866	1.219	1.177–1.263
CAD	0.963	0.770–1.205	1.006	0.979–1.034
Arrhythmia	1.315	1.010–1.713	1.428	1.384–1.474
Hyperlipidemia	0.885	0.712–1.101	0.806	0.785–0.827
Stroke	1.216	0.987–1.498	1.312	1.279–1.346
Arthritis	1.039	0.824–1.311	0.853	0.831–0.877
Asthma	0.936	0.749–1.171	1.090	1.062–1.118
Cancer	2.846	2.310–3.506	2.547	2.484–2.611
CKD	1.453	0.984–2.146	1.361	1.303–1.423
COPD	0.934	0.703–1.241	1.320	1.281–1.361
Diabetes	1.079	0.869–1.34	1.254	1.221–1.289
Chronic hepatitis	1.343	0.648–2.787	1.128	1.02–1.247
Liver cirrhosis	2.365	1.472–3.800	2.196	2.084–2.315
Osteoporosis	0.639	0.505–0.808	0.918	0.893–0.944

aHR: adjusted hazard ratio; CI: confidence interval; CHF: congestive heart failure; CAD: coronary artery disease; CKD: chronic kidney disease; COPD: chronic obstructive pulmonary disease.

**Table 4 ijerph-19-09989-t004:** Association of mental disorder and risk of all-cause mortality among people with chronic diseases.

	Risk of All-Cause Mortality	
	aHR	95% CI
Hypertension	1.195	1.160–1.230
CHF	1.222	1.180–1.265
CAD	1.005	0.978–1.033
Arrhythmia	1.427	1.383–1.472
Hyperlipidemia	0.807	0.786–0.829
Stroke	1.310	1.277–1.344
Arthritis	0.856	0.834–0.880
Asthma	1.088	1.060–1.116
Cancer	2.550	2.488–2.614
CKD	1.363	1.304–1.423
COPD	1.313	1.274–1.353
Diabetes	1.251	1.218–1.285
Chronic hepatitis	1.131	1.024–1.250
Liver cirrhosis	2.198	2.086–2.316
Osteoporosis	0.913	0.888–0.939

aHR: adjusted hazard ratio; CI: confidence interval; CHF: congestive heart failure; CAD: coronary artery disease; CKD: chronic kidney disease; COPD: chronic obstructive pulmonary disease.

**Table 5 ijerph-19-09989-t005:** Comparing 3 groups: sociodemographic and clinical characteristics.

Variables	People with History of Psychiatric Hospitalization (n = 404)	People with History of Psychiatric Outpatient (n = 6795)	People without Mental Illness (n = 485,965)	*p*-Value
**Gender**				<0.0001
Male	196 (48.51)	2999 (48.51)	230,477 (53.6)	
Female	208 (51.49)	3796 (51.49)	225,488 (46.4)	
**Age**				<0.0001
40–49	169 (41.83)	2910 (42.83)	225,318 (46.37)	
50–59	119 (29.46)	2061 (30.33)	141,786 (29.18)	
60–69	76 (18.81)	1432 (21.07)	93,447 (19.23)	
>70	40 (9.90)	392 (5.77)	25,414 (5.23)	
**Income level**				0.0689
0–3 decile	112 (27.72)	1564 (23.02)	109,233 (22.48)	
4–7 decile	116 (28.71)	2225 (32.74)	158,124 (32.54)	
8–10 decile	176 (43.56)	3006 (44.24)	218,608 (44.98)	
**Number of health screenings**				<0.0001
≥5	86 (21.29)	3001 (44.10)	227,135 (46.74)	
4	61 (15.10)	1429 (21.03)	95,829 (19.72)	
3	49 (12.13)	753 (11.08)	52,021 (10.70)	
2	51 (12.62)	526 (7.74)	37,392 (7.69)	
1	66 (16.34)	495 (7.28)	33,689 (6.93)	
0	91 (22.52)	591 (8.7)	39,899 (8.21)	
**Physical activity**				0.1127
Good	187 (28.15)	2735 (33.91)	193,335 (33.82)	
Poor	217 (71.85)	4060 (66.09)	292,630 (66.18)	
**Smoking**				<0.0001
Never smoking	145 (46.33)	3079 (49.66)	203,167 (45.56)	
Ex smoking	59 (18.85)	1031 (16.63)	91,585 (20.54)	
Current smoking	109 (34.82)	2090 (33.71)	15,1148 (33.9)	
**Alcohol drinking**				<0.0001
None	40 (12.82)	524 (8.45)	32,014 (7.18)	
<2 times/week	75 (24.04)	1465 (23.62)	123,539 (27.7)	
≥2 times/week	197 (65.12)	4213 (63.14)	290,373 (65.12)	
**Blood pressure (mmHg)**				0.0009
Normal	109 (34.82)	1845 (29.74)	125,334 (28.1)	
Prehypertension	161 (51.44)	3656 (58.93)	267,056 (59.87)	
Hypertension	43 (13.74)	703 (11.33)	53,676 (12.03)	
**Total cholesterol (mg/dL)**				0.0002
<200	230 (73.48)	4606 (74.24)	337,862 (77.25)	
200–239	56 (17.89)	1298 (20.92)	88,620 (13.33)	
≥240	27 (8.63)	300 (4.84)	19,584 (4.39)	
**Fasting glucose level (mg/dL)**				<0.0001
<100	159 (50.8)	3531 (56.91)	249,950 (56.03)	
100–125	117 (37.38)	2108 (33.98)	146,805 (32.91)	
≥126	37 (11.82)	565 (9.11)	49,311 (11.05)	

Values are presented as numbers (%).

**Table 6 ijerph-19-09989-t006:** Comparing 3 groups: prevalence of chronic diseases.

Chronic Diseases	People with History of Psychiatric Hospitalization (n = 404)	People with History of Psychiatric Outpatient (n = 6795)	People without Mental Illness (n = 485,965)	*p*-Value
Hypertension	230 (56.93)	3518 (51.77)	240,006 (49.39)	<0.0001
CHF	29 (7.18)	359 (5.28)	23,827 (4.90)	0.0382
CAD	92 (22.77)	1206 (17.75)	78,494 (16.15)	<0.0001
Arrhythmia	40 (9.9)	517 (7.61)	32,212 (6.63)	0.0002
Hyperlipidemia	201 (49.75)	3654 (53.77)	240,957 (49.58)	<0.0001
Stroke	97 (24.01)	1176 (17.31)	69,925 (14.39)	<0.0001
Arthritis	234 (57.92)	4665 (68.65)	309,464 (63.68)	<0.0001
Asthma	99 (24.50)	1542 (22.69)	101,393 (20.86)	0.0002
Cancer	48 (11.88)	697 (10.26)	52,114 (10.72)	0.3610
CKD	15 (3.71)	127 (1.87)	10,303 (2.12)	0.0302
COPD	39 (9.65)	502 (7.39)	33,366 (6.87)	0.0210
Diabetes	170 (42.08)	2090 (30.76)	148,171 (30.49)	<0.0001
Chronic hepatitis	6 (1.49)	56 (0.82)	3691 (0.76)	0.2041
Liver cirrhosis	10 (2.48)	69 (1.02)	6086 (1.25)	0.0187
Osteoporosis	89 (22.03)	1816 (26.73)	106,007 (21.81)	<0.0001

Values are presented as numbers (%). CHF: congestive heart failure; CAD: coronary artery disease; CKD: chronic kidney disease; COPD: chronic obstructive pulmonary disease.

**Table 7 ijerph-19-09989-t007:** Comparing 3 groups: association of chronic diseases and risk of all-cause mortality.

Chronic Diseases	People with History of Psychiatric Hospitalization (n = 404)		People with History of Psychiatric Outpatient (n = 6795)		People without Mental Illness (n = 485,965)	
	aHR	95% CI	aHR	95% CI	aHR	95% CI
Hypertension	2.406	0.815–7.107	1.315	1.016–1.702	1.192	1.158–1.228
CHF	0.771	0.190–3.136	1.505	1.116–2.030	1.219	1.117–1.263
CAD	2.889	1.117–7.474	0.908	0.718–1.149	1.006	0.979–1.034
Arrhythmia	1.143	0.310–4.220	1.401	1.064–1.846	1.428	1.384–1.474
Hyperlipidemia	0.378	0.150–0.955	0.983	0.781–1.237	0.806	0.785–0.827
Stroke	1.294	0.495–3.378	1.259	1.014–1.563	1.312	1.279–1.346
Arthritis	0.578	0.204–1.640	1.143	0.892–1.465	0.853	0.931–0.877
Asthma	0.742	0.285–1.933	0.917	0.725–1.158	1.090	1.062–1.118
Cancer	4.094	1.474–11.373	2.820	2.270–3.502	2.547	2.484–2.611
CKD	0.545	0.069–4.331	1.630	1.078–2.465	1.361	1.303–1.423
COPD	0.725	0.185–2.838	1.010	0.756–1.350	1.320	1.281–1.361
Diabetes	1.461	0.565–3.778	1.022	0.813–1.284	1.254	1.221–1.289
Chronic hepatitis	NA	NA	1.737	0.835–3.614	1.128	1.020–1.247
Liver cirrhosis	26.334	3.685–188.178	2.160	1.291–3.613	2.196	2.084–2.315
Osteoporosis	0.143	0.039–0.515	0.675	0.529–0.861	0.918	0.893–0.944

aHR: adjusted hazard ratio; CI: confidence interval; CHF: congestive heart failure; CAD: coronary artery disease; CKD: chronic kidney disease; COPD: chronic obstructive pulmonary disease.

## Data Availability

The Korean National Health Insurance Service–National Sample Cohort is a public, open-access database. It is based on the health insurance claim data of all Koreans, and the sample cohort is available for public purposes and scientific research. The authors do not have permission to share these data. The sample cohort data are available after acceptance of approval for use by the national health insurance service. (https://nhiss.nhis.or.kr/bd/ab/bdaba000eng.do).

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
