# Peer review of "Effects of Chronic Diseases on All-Cause Mortality in People with Mental Illness: A Retrospective Cohort Study Using the Korean National Health Insurance Service-Health Screening"

_ijerph, 2022, doi:10.3390/ijerph19169989_

Round 1

Reviewer 1 Report

The work presented contains inaccuracies that need to be clarified. Please clarify:

1. whether the unequal gender distribution (female/male) in the study groups can affect the frequency of chronic diseases found, such as asthma, osteoporosis, COPD?

2. the data in Table 4 differs from the data in the description, why? And which data is correct?

3. Can it affect the conclusions presented?

Author Response

Response to Reviewer 1 Comments

Thank you for giving me the opportunity to submit a revised draft of my manuscript titled "Effects of Chronic Diseases on All-Cause Mortality in People with Mental Illness: A Retrospective Cohort Study Using the Korean National Health Insurance Service-Health Screening". We appreciate the time and effort that you have dedicated to providing your valuable feedback on my manuscript. We are grateful to you for insightful comments on my paper. We have been able to incorporate changes to reflect most of the suggestions.

Point 1: whether the unequal gender distribution (female/male) in the study groups can affect the frequency of chronic diseases found, such as asthma, osteoporosis, COPD?

Response 1: We fully agree with your opinion. Because we used raw data, the unequal gender distribution between people with mental illness and people without mental illness is likely to have affected the prevalence of chronic diseases such as asthma, osteoporosis, and COPD. Furthermore, the unequal age distribution is also is likely to have affected the mortality. The problem could be solved by controlling gender and age through propensity score matching, but it was not conducted due to the disadvantage of reducing the overall sample size. We will include this as a limitation (page 13, line 364-367).

Point 2: the data in Table 4 differs from the data in the description, why? And which data is correct?

Response 2: We discovered that there was a mistake in copying the results into Table 4. We modified the contents of Table 4 (page 8, line 219-233) according to the real results. We apologize for confusion caused by our mistake to you.

Point 3: Can it affect the conclusions presented?

Response 3: Since the contents of the manuscript were prepared based on the real results, there was no effect on the conclusions.

The revised manuscript file is attached. Please see the attachment.

Thank you .

Reviewer 2 Report

The authors used a cohort collected from a participants of the National Screening Program during 2002-2003, and followed up from 2010 to 2019, to estimate the difference in disease burden and mortality between those with mental illnesses and those without. People with mental illnesses was defined as those who had a record of being treated for a mental illness listed with an F code between 2002 and 2009. The reviewer has the following concerns about this study:

1. The status of mental illness: (a) It was not stated whether mental illness occurring before 2002-2003 was recorded, and how that piece of information was used to classify persons with/without mental illness. (b) It was not described in the group of mental illness, at which age were they given the first diagnosis. (c) The authors did not describe the distribution of diagnoses of mental illnesses. (d) On the status of confirmed diagnosis and severity of mental illness, the authors did not differentiate between those with long-term illness/treatment and those with sporadic outpatient visits. 

2. If the issues above were not addressed, then the authors could face problems of mis-classification, because their mental illness before 40 was unknown, and because the mental illness group may contain unconfirmed cases.

3. Given the method of recruitment, the authors could have a cohort of people mental illnesses that had particular characteristics which makes comparison with other studies difficult. For example, (a) it is from active participation in national screening programs, whose characteristics may be different from those who do not participate. (b) it looks like a cohort of late onset age (more than 40 years old), (c) it looks like a cohort of minor mental illness (judged by the fact that 63% of them had active participation in health screenings). 

4. For figure 1, the cohort started with a wide age range, but it is not clear how the age factor was treated in this analysis. The graph doesn't show that the two lines had 8.9% difference. 

5. For table 1, it is not clear when data about the characteristics of the two groups were collected.

6. For table 2, it is not clear which factors were controlled for: age, sex, smoking, drinking, physical activity, income?

7. For table 3 and 4, when the authors did the analysis conditional on the presence of particular chronic diseases, the risk of mortality because of (if increased) occurrence of these could not be assessed.

Based on these concerns, the merit of the data collected could not be fully presented by the results. 

Author Response

Response to Reviewer 2 Comments

Thank you for giving me the opportunity to submit a revised draft of my manuscript titled "Effects of Chronic Diseases on All-Cause Mortality in People with Mental Illness: A Retrospective Cohort Study Using the Korean National Health Insurance Service-Health Screening". We appreciate the time and effort that you have dedicated to providing your valuable feedback on my manuscript. We are grateful to you for insightful comments on my paper. We have been able to incorporate changes to reflect most of the suggestions. 

Point 1: The authors used a cohort collected from a participants of the National Screening Program during 2002-2003, and followed up from 2010 to 2019, to estimate the difference in disease burden and mortality between those with mental illnesses and those without. People with mental illnesses was defined as those who had a record of being treated for a mental illness listed with an F code between 2002 and 2009. The reviewer has the following concerns about this study: The status of mental illness

Point 1(a): It was not stated whether mental illness occurring before 2002-2003 was recorded, and how that piece of information was used to classify persons with/without mental illness.

Response 1(a): Since the Korean National Health Insurance Service-Health Screening Data was established between 2002 and 2009, psychiatric medical records before 2002 were unknown. Therefore, psychiatric medical records prior to 2002 were not used to classify persons with/without mental illness (page 13, line 359-362).

Point 1(b): It was not described in the group of mental illness, at which age were they given the first diagnosis.

Response 1(b): Since the sample data includes medical records from 2002 to 2019, if the first diagnosis was made before 2002, the age at the first diagnosis was not known. However, many mental disorders are first diagnosed before the age of 40, we did not consider the first diagnosis (page 2, line 94 ~ page 3, line 95-96). If people with mental illness were defined as people diagnosed with mental illness for the first time from 2002 to 2009, as pointed out, there is a high possibility that the mental illness onset after the age of 40 will be biased.

Point 1(c): The authors did not describe the distribution of diagnoses of mental illnesses.

Response 1(c): We defined people with mental illness as those with ICD-10 F-code diagnoses. Regarding the sample data from the Korean National Health Insurance Service, the F code was a sensitive diagnosis code, and it was forbidden to indicate the detailed diagnosis name; therefore, it was not possible to describe the distribution of diagnoses of mental illnesses. We included this as a limitation (page 13, line 351-356).

Point 1(d): On the status of confirmed diagnosis and severity of mental illness, the authors did not differentiate between those with long-term illness/treatment and those with sporadic outpatient visits.

Response 1(d): Although we did not investigate the frequency of treatment in people with mental illness, we did try to differentiate severity by whether or not they were hospitalized. We divided the participants into people with history of psychiatric hospitalization, people with history of psychiatric outpatient only and people without mental illness. Sub-analysis content has been added to Table5,6,7. To summarize the results, number of people with history of psychiatric hospitalization was 404 (0.08%) and people with history of psychiatric outpatient only was 6,795 (1.38%). The all-cause mortality of people with history of psychiatric hospitalization was higher than those of people with history of psychiatric outpatient only and those of people without mental illness (25.00% vs. 10.60% vs. 10.28%, p<0.0001) (page 8, line 225-230). There was a significant difference in the prevalence of chronic diseases among the psychiatric inpatient group, the psychiatric outpatient only group, and people without mental illness (Table 6) (page 10, line 240-252). In addition, the psychiatric inpatient group showed a higher aHR than the psychiatric outpatient only group and people without mental illness in cancer (aHR 4.094, 95% CI 1.474-11.373 vs. aHR 2.820, 95% CI 2.270-3.502 vs. aHR 2.547, 95% CI 2.484-2.611) and liver cirrhosis (aHR 26.334 95% CI 3.685-188.178 vs. aHR 2.160, 95% CI 1.291-3.613 vs. aHR 2.196, 95% CI 2.084-2.315)(Table 7) (page 10, line 254-256 ~ page 11, line 257 - 264).

Point 2: If the issues above were not addressed, then the authors could face problems of mis-classification, because their mental illness before 40 was unknown, and because the mental illness group may contain unconfirmed cases.

Response 2: You have raised an important point here. However, we regardless of the presence or absence of a psychiatric diagnosis before 2002, people who had psychiatric medical records between 2002 and 2009, were defined as people with mental illness(page 2, line 94 ~ page 3, line 95-96). Therefore, people who diagnosed before 2002 but had no psychiatric medical records between 2002 and 2009 were not included in people with mental illness. We included this as a limitation(page 13, line 359-362).

Point 3: Given the method of recruitment, the authors could have a cohort of people mental illnesses that had particular characteristics which makes comparison with other studies difficult. For example,

(a): it is from active participation in national screening programs, whose characteristics may be different from those who do not participate.

(b) it looks like a cohort of late onset age (more than 40 years old),

(c) it looks like a cohort of minor mental illness (judged by the fact that 63% of them had active participation in health screenings).

Response 3(a),(c): I completely agree with your opinion. However, total participants was a randomized 10% sample from among those subject to health screening, including those who had never taken a health screening. Among them, we selected people with mental illness. As pointed out in 3(C), the reason that the health screening rate of people with mental illness is higher than in other studies is that it includes many people with mild mental illness. The proportion of active participants who had health screenings 4 or more times in 10 years was about 63% for those with mental illness and about 65% for people without mental illness. However, from 2002 to 2009, among people with a history of psychiatric hospitalization, only 36% of people with mental illness were active participants (Table 5) (page 8, line 231-238 ~ page 9, 239).

Response 3(b): We did not consider age at first diagnosis. Since people with mental illness include those with a first diagnosis before 2002, they are not biased toward late age onset mental illness (page 2, line 94 ~ page 3, line 95-96).

Point 4: For figure 1, the cohort started with a wide age range, but it is not clear how the age factor was treated in this analysis. The graph doesn't show that the two lines had 8.9% difference.

Response 4: Age factor is not included. The follow up years of the K-M graph was prepared using raw data on the time of death or the end of observation.

Point 5: For table 1, it is not clear when data about the characteristics of the two groups were collected.

Response 5: Gender, age, and income level were based on 2002 (page 3, line 100-101). The number of health screenings was the total number of health screenings between 2002-2009 (page 3, line 105-106). Physical activity, smoking, and alcohol drinking were based on the last health screenings during 2002-2009. Blood pressure, total cholesterol, and fasting blood glucose were averages from 2002-2009 (page 3, line 114-115).

Point 6: For table 2, it is not clear which factors were controlled for: age, sex, smoking, drinking, physical activity, income?

Response 6: All Cox proportional hazard models were fully adjusted with the covariates presented in Table  1

Point 7: For table 3 and 4, when the authors did the analysis conditional on the presence of particular chronic diseases, the risk of mortality because of (if increased) occurrence of these could not be assessed.

Response 7: I completely agree with your opinion. The presence or absence of chronic disease was also judged by the presence or absence of medical records from 2002 to 2009, so it was not considered whether the record was the first diagnosis, that is, a new onset.

In addition, we discovered that there was a mistake in copying the results into Table 4. We modified the contents of Table 4 according to the real results (page 8, line 219-222). We apologize for confusion caused by our mistake to you. Since the contents of the manuscript were prepared based on the real results, however, there was no effect on the conclusions.

The revised manuscript file is attached. Please see the attachment.

Thank you .

Reviewer 3 Report

The article is well written, following the recommended structure. The information is clear and relatively easy to understand. Another strength is the impressive number of patients from whom the data was collected.

The topic addressed is one of interest, with significant implications for medical practice. The sample is a large one, but with significant differences between the two groups of patients. The fact that specific mental disorder was not indicated is a minus that can affect the interpretation of the results and the identification of more specific implications for medical practice. The authors do not indicate what implications these results have for practice? What clinical interventions they propose, and what suggestions they have for future research.

It is also recommended that they better emphasize what is the aim of the study, its relevance and the added knowledge in the introduction.

Author Response

Response to Reviewer 3 Comments

Thank you for giving me the opportunity to submit a revised draft of my manuscript titled "Effects of Chronic Diseases on All-Cause Mortality in People with Mental Illness: A Retrospective Cohort Study Using the Korean National Health Insurance Service-Health Screening". We appreciate the time and effort that you have dedicated to providing your valuable feedback on my manuscript. We are grateful to you for insightful comments on my paper. We have been able to incorporate changes to reflect most of the suggestions. We have highlighted the changes within the manuscript. The revised manuscript file is attached. Please see the attachment.

Thank you.

Round 2

Reviewer 2 Report

The authors have tried to answer most of the comment in the first round of the review and improved accordingly, but the credit of the study is limited by the availability of data, esp. the diagnosis of mental illness, their severity, age of onset, and treatment. In addition, two possibilities cannot be ruled out: (1) the misclassification of cases and controls is inevitable since the authors do not have record prior to 2002, and (2) the exclusion of more serious cases of mental illness who were less likely to participate in a voluntary health screening. These are barriers that could not be addressed by post-hoc data analyses. 

Author Response

Response to Reviewer 2 Comments

Thank you for giving me the opportunity to submit a revised draft of my manuscript titled "Effects of Chronic Diseases on All-Cause Mortality in People with Mental Illness: A Retrospective Cohort Study Using the Korean National Health Insurance Service-Health Screening". We have been able to incorporate changes to reflect most of the suggestion.

Point 1: The authors have tried to answer most of the comment in the first round of the review and improved accordingly, but the credit of the study is limited by the availability of data, esp. the diagnosis of mental illness, their severity, age of onset, and treatment. In addition, two possibilities cannot be ruled out: (1) the misclassification of cases and controls is inevitable since the authors do not have record prior to 2002, and (2) the exclusion of more serious cases of mental illness who were less likely to participate in a voluntary health screening. These are barriers that could not be addressed by post-hoc data analyses.

Response 1: I totally agree with your opinion. There was clear limitations because of the data of this study from the KNHIS-HEALS cohort database. In the KNHIS-HEALS cohort database, mental illness is classified as a sensitive disease, so detailed diagnosis names are not indicated and all are indicated as F***. As a result, it was not possible to describe the distribution of diagnoses of mental disorders. In addition, under the age of 40, only householder of locally provided policyholders or employer-provided policyholders are eligible for subjects of the national health screening program. As a result, participants under the age of 40 were not included in this study, because the KNHIS-HEALS cohort database is sampled with subjects over the age of 40, which is the age of when all the Koreans are subjects to national health screening program.

Response 1(1): Furthermore, the KNHIS-HEALS cohort database did not include medical records before 2002 and information on the date of first diagnosis. As a results, there was a possibility that misclassification of people with mental illness because people who diagnosed before 2002 but had no psychiatric record between 2002 and 2009 were not included in people with mental illness. Despite such disadvantages, we used the KNHIS-HEALS cohort database because the data included information such as physical activity, smoking, alcohol drinking, blood pressure, fasting glucose level, etc. Since the risk factors could influence the mortality, we controlled the risk variables and presented an association between chronic diseases and the risk of all-cause mortality. Future studies to investigate the effects of chronic diseases on all-cause mortality for each psychiatric disorder are proposed. In addition, participants should not be affected by the age of onset for mental disorders, including all ages, and the date of initial diagnosis should be included. We included this as limitations.

Response 1(2): People with mental illness who were less likely to participate in a voluntary health screening were also included in participants. In table 1, a total of 682 (9.47%) of people with mental illness who have never had a health screening, and a total of 39,899 (8.21%) of those without mental illness who have never had a health screening are presented.